# Dupilumab Improves Skin Barrier Function in Adults with Atopic Dermatitis: A Prospective Observational Study

**DOI:** 10.3390/jcm11123341

**Published:** 2022-06-10

**Authors:** Trinidad Montero-Vilchez, Juan-Angel Rodriguez-Pozo, Pablo Diaz-Calvillo, Maria Salazar-Nievas, Jesús Tercedor-Sanchez, Alejandro Molina-Leyva, Salvador Arias-Santiago

**Affiliations:** 1Department of Dermatology, Virgen de las Nieves University Hospital, 18012 Granada, Spain; tmonterov@gmail.com (T.M.-V.); juanangelrpg@gmail.com (J.-A.R.-P.); pdc.muro@gmail.com (P.D.-C.); msalazarn@hotmail.com (M.S.-N.); jesustercedor@gmail.com (J.T.-S.); salvadorarias@hotmail.es (S.A.-S.); 2Biosanitary Institute of Granada (ibs.GRANADA), 18012 Granada, Spain; 3Cell Production and Tissue Engineering Unit, Virgen de las Nieves University Hospital, Andalusian Network of Design and Translation of Advanced Therapies, 18012 Granada, Spain; 4Department of Dermatology, Faculty of Medicine, University of Granada, 18071 Granada, Spain

**Keywords:** atopic dermatitis, dupilumab, skin barrier, transepidermal water loss

## Abstract

Epidermal barrier dysfunction plays an important role in atopic dermatitis (AD). The difficulty of objectively assessing AD severity and the introduction of new biologicals into clinical practice highlight the need to find parameters to monitor clinical outcomes. The aim of this study is to evaluate the impact of dupilumab on skin barrier function and compare it with other treatments in patients with AD. A prospective observational study was conducted in adults with AD treated with topical corticosteroids (TCS), cyclosporine, or dupilumab. The main outcome measures after 16 weeks of treatment were Eczema Area and Severity (EASI)-50 (50% improvement in EASI), and transepidermal water loss (TEWL)-50 (50% improvement in TEWL). Forty-six patients with AD were included in the study. The proportion of patients who achieved EASI-50 at week 16 was significantly higher in patients receiving dupilumab (81.8% vs. 28.6% vs. 40%, *p* = 0.004). In eczematous lesions, TEWL decreased in patients receiving dupilumab (31.02 vs. 12.10 g·h^−1^·m^−2^, *p* < 0.001) and TCS (25.30 vs. 14.88 g·h^−1^·m^−2^, *p* = 0.047). The proportion of patients who achieved TEWL-50 at week 16 was higher for dupilumab than for cyclosporine or TCS. Temperature only decreased in the dupilumab group. Stratum corneum hydration increased in eczematous lesions and non-involved skin only in patients with dupilumab. In conclusion, dupilumab improves skin barrier function in patients with AD better than TCS or cyclosporine, both in eczematous lesions and in non-lesioned skin.

## 1. Introduction

Atopic dermatitis (AD) is a chronic cutaneous inflammatory disease caused by genetic and environmental factors [1,2]. It is one of the most prevalent skin diseases, with a prevalence ranging from 0.96% to 22.6% in children and from 1.2% to 17.1% in adults [3], with higher prevalence in industrialized countries [4]. Clinically, it is characterized by recurrent and itchy eczematous lesions, excoriations, scaling and dry skin [1,5]. It is also a disease that greatly impairs the quality of life of patients and their cohabitants [6].

Epidermal barrier dysfunction, immune dysregulation, and gut dysbiosis may play a role in this disease [7]. Skin barrier dysfunction is considered the first step in the development of AD [8,9]. Filaggrin (FLG) mutations lead to alterations in the differentiation and growth of a normal stratum corneum, increasing transepidermal water loss (TEWL) [10]. Moreover, skin barrier dysfunction increases allergic sensitization to antigens [11], an independent risk factor for developing food sensitization [12]. Skin barrier impairment, reflected in high TEWL and temperatures is also related to more severe disease [13]. 

Multiple diagnostic tools are used to evaluate severity in patients with AD, most commonly the Eczema Area and Severity Index (EASI) and SCORing Atopic Dermatitis (SCORAD) [14,15]. The EASI is a medical evaluation of extension and intensity, while the SCORAD also includes a patient assessment of itch and sleeplessness [1,14]; both these parameters have a subjective component that could lead to a high intra- and interobserver variability [16,17,18].

Available therapies for AD include topical corticosteroids (TCS), calcineurin inhibitors, phototherapy and systemic immunotherapies, such as cyclosporine [1]. Dupilumab, a fully human monoclonal antibody that binds specifically to the shared α chain subunit of the interleukin-4 and interleukin-13 receptors, is associated with clinical improvement in patients with AD, reducing EASI, SCORAD and Dermatology Life Quality Index (DLQI) with an acceptable safety profile [19]. Other biologics (dupilumab, tralokinumab) and JAK-inhibitors are being developed that are showing similar or even better score outcomes [20,21]. Nevertheless, there is scant information about how these treatments modify skin barrier function [7].

The difficulty in objectively assessing AD severity and the introduction of new biologicals in clinical practice highlight the need to find parameters to select the most appropriate treatment and for monitoring outcomes [22,23]. Skin barrier function parameters are easy to evaluate objectively [24,25] and may be suitable tools for assessing the efficacy of AD therapies.

Thus, the aim of this study is to evaluate the impact of dupilumab on skin barrier function and compare it with other treatments in patients with AD.

## 2. Materials and Methods

### 2.1. Study Design and Participants

We conducted a prospective observational study in participants recruited from September 2019 to May 2021, in the Department of Dermatology, Virgen de las Nieves University Hospital, Granada, Spain.

Eligible patients were adults diagnosed with AD by a dermatologist according to Hanifin and Rajka criteria [26,27], 18–65 years of age, with an eczematous lesion on the volar forearm, who were scheduled to start a new treatment with TCS, cyclosporine or dupilumab. Only patients with an eczematous lesion on the volar forearm were included because the measurements were always taken in this area to homogenize differences in skin barrier function that may exist in different body areas [25]. The exclusion criteria were: a clinical infection on the measured area; history of cancer; immunological disease or other inflammatory skin disease; incapacity to comply with the study protocol and no signed informed consent form.

Patients were assigned to treatment with TCS, cyclosporine or dupilumab following current clinical recommendations for treating AD [1]. Mild-to-moderate patients who had not received treatment for at least the previous six months were treated with mometasone cream once daily (TCS group). Moderate patients who had not responded to TCS and severe patients were assigned to the cyclosporine group and advised to avoid any topical treatment for one week prior to baseline evaluation. Cyclosporine was initiated at a high dose, 5 mg/kg/day, for a 3–6-week induction phase, followed by gradual tapering of the dose based on clinical response to a dose of 2–3 mg/kg/day in the maintenance phase. Moderate to severe patients who did not respond to cyclosporine or had some contraindication to receiving it were assigned to the dupilumab group. Dupilumab 300 mg was administered subcutaneously every other week following a loading dose of 600 mg. Immunosuppressive agents were discontinued at least 4 weeks prior to starting dupilumab in all patients. Patients receiving cyclosporine and dupilumab were allowed to use TCS twice a week if needed. Both patients starting cyclosporine and dupilumab should not use TCS or calcineurin inhibitors at least 24 h before baseline measurement. Patients in all groups were allowed to use emollients and moisturizers when needed but not 24 h before the baseline measure.

### 2.2. Outcomes and Measures

The main outcome measure to assess clinical improvement was EASI-50 (50% improvement in EASI) at treatment week 16 [28], and the primary outcome measure to assess skin barrier improvement was TEWL-50 (50% improvement in TEWL) at treatment week 16. 

#### 2.2.1. Clinical Assessment

AD severity was assessed by EASI, SCORAD, DLQI, the Investigator Global Assessment (IGA) scale and body surface area (BSA). EASI is calculated by independently assessing body surface involvement in four body regions (head and neck, upper extremities, trunk and lower extremities) and evaluating erythema, induration/papulation/edema, excoriations, and lichenification in each area [29]. SCORAD consists of the evaluation of disease extension, intensity (composed of six items: erythema, edema/papules, effect of scratching, oozing/crust formation, lichenification and dryness) and subjective symptoms (itch, sleeplessness) [15,27]. DLQI evaluates the impact of dermatological conditions and consists of a 10-item questionnaire addressing patient quality of life [30]. All clinical indexes were determined by a dermatologist at baseline and after the 16-week follow-up.

#### 2.2.2. Skin Barrier Function Parameters

TEWL (in g·h^−1^·m^−2^), skin temperature (in °C), stratum corneum hydration (SCH) (in arbitrary units, AU), and pH were respectively measured using the Tewameter TM 300, Skin-Thermometer ST 500, Corneometer CM 825, and Skin-pH-Meter PH 905, connected to a Multi Probe Adapter (MPA, Courage + Khazaka electronic GmbH, Bilbao, Spain). All parameters were evaluated on an eczematous area on the volar forearm and on a non-involved area 5 cm from the affected area, and the average value from ten measurements in each location was used for analysis. Measurements were taken at baseline and at the 16-week follow-up after resting at least for 30 min in a room with controlled ambient air temperature and humidity, measured with the TFA Lab Thermometer IP65 LT-101, Wertheim, Germany (average air temperature 22 ± 1 °C; ambient air humidity of 45% ± 5%). No systemic or topical treatments were allowed in the six hours before the measurements were taken.

#### 2.2.3. Other Variables

Age, sex, smoking/alcohol habits, marital status, education level, skin hydration habit, family history of AD, age of disease onset, signs of atopy march, and body mass index (BMI) were recorded by means of a clinical interview and physical examination. Regarding skin hydration habit patients were asked about the frequency they used emollients per week and were classified into those applying them ≤4 or >4 times per week.

### 2.3. Statistical Analysis

Descriptive statistics were used to present the sample characteristics. Continuous data were expressed as the mean (standard deviation) and qualitative data as relative (absolute) frequency. The Shapiro–Wilk test was used to determine the normality of data distribution and Levene’s test to check the homogeneity of variance. One-way analysis of variance (ANOVA) was used to compare quantitative variables between different treatments. The Student’s *t*-test for paired samples was used to compare differences in parameters before and after using the hand hygiene product. A multivariable logistic regression model was constructed to evaluate variables associated with TEWL-50 and EASI-50. Epidemiological and statistical criteria were used to model variable selection. The effect of each exploratory variable on the model and its significance were studied. If the variable improved the model fit and adequacy (based on the likelihood ratio criteria and the significance of the parameter), it was kept; otherwise, the variable was excluded. The model was checked for pairwise interaction between covariates. Potential confounding covariates were studied using a change of significance in the model’s parameters or a change of 30% of its value. Statistical significance was defined as a two-tailed *p* < 0.05. SPSS version 24.0 (SPSS Inc, Chicago, IL, USA) was used for statistical analyses.

### 2.4. Ethics

This study was approved by the ethics committee of Hospital Universitario Virgen de las Nieves (HC01/0442-N-20, Impact of topical, systemic or physical treatment on skin homeostasis in patients with skin diseases). The nature of the study was explained to all participants, who agreed to participate by giving their verbal and written consent. All measurements were non-invasive, and the confidentiality of participants’ data was strictly preserved.

## 3. Results

### 3.1. Baseline Demographic and Clinical Characteristics

Sixty-two individuals were assessed for eligibility, and 46 patients with AD were finally included, 10 of whom were treated with TCS, 14 with cyclosporine and 22 with dupilumab. Demographic and clinical characteristics are shown in Table 1. Patients receiving TCS were younger than those treated with dupilumab and cyclosporine (19.90 vs. 28.95 vs. 35.64 years, *p* = 0.042). Patients in the TCS group were more frequently single than those in the dupilumab and cyclosporine groups (100% vs. 81.8% vs. 63.3%, *p* = 0.036). Dupilumab patients had a longer disease duration than those receiving cyclosporine or TCS (20.67 vs. 8.38 vs. 12.20, *p* = 0.004). After follow-up, 8.7% (4/46) of the patients did not complete the study (1 TCS, 2 cyclosporine, 1 dupilumab), Appendix A.

### 3.2. Clinical Improvement

At baseline, patients who received dupilumab had more severe disease than those who received cyclosporine and TCS, reflected in higher EASI (24.60 vs. 15.04 vs. 11.72, *p* < 0.001), SCORAD (57.30 vs. 47.41 vs. 37.28, *p* < 0.001), DLQI (17.59 vs. 9.29 vs. 12.20, *p* < 0.001), BSA (39.54 vs. 22.63 vs. 14.23, *p* < 0.001) and IGA scores (3.73 vs. 3.21 vs. 2.5, *p* < 0.001), Table 1.

The proportion of patients who achieved EASI-50 at week 16 was significantly higher among patients receiving dupilumab than those receiving cyclosporine or topical corticosteroids (81.8% vs. 28.6% vs. 40%, *p* = 0.004), Figure 1. After conducting a multivariate logistic regression model adjusted by age, age of disease onset, sex, smoking habit and skin hydration habit, treatment with dupilumab emerged as an independent factor for achieving EASI-50 (OR = 10.67, *p* = 0.026). 

Patients receiving dupilumab also showed greater improvement in SCORAD (−29.26 vs. −15.69 vs. −12.79, *p* < 0.001), DLQI (−12.52 vs. −2.56 vs. −3.75, *p* < 0.001), BSA and IGA scores than those treated with cyclosporine and TCS (Figure 2, Appendix A). 

### 3.3. Skin Barrier Function

Skin barrier function parameters did not differ at baseline between groups, see Appendix A. In eczematous lesions, TEWL decreased in patients receiving dupilumab (31.02 vs. 12.10 g·h^−1^·m^−2^, *p* < 0.001) and TCS (25.30 vs. 14.88 g·h^−1^·m^−2^, *p* = 0.047) but did not change with cyclosporine. Temperature only decreased in the dupilumab group (32.53 vs. 31.64 °C, *p* = 0.009). SCH increased in patients treated with dupilumab (19.93 vs. 37.73 AU, *p* < 0.001) and TCS (18.24 vs. 40.79 AU, *p* = 0.010), but did not change with cyclosporine. pH did not change in any group. In non-lesioned skin, TEWL (11.87 vs. 8.25 g·h^−1^·m^−2^, *p* = 0.006) and SCH (32.68 vs. 41.68 AU, *p* < 0.001) only improved in patients receiving dupilumab. Temperature, SCH and pH did not change in non-lesioned skin after receiving any treatment, see Appendix A. 

The proportion of patients who achieved TEWL-50 at week 16 was greater for patients receiving dupilumab than for those receiving cyclosporine or topical corticosteroids (50% vs. 14.3% vs. 30%, *p* = 0.101). Furthermore, after a multivariate logistic regression model adjusted for age, age of disease onset, sex, smoking habit and skin hydration habit, dupilumab treatment with dupilumab emerged as an independent factor for achieving TEWL-50 (*p* = 0.004), see Figure 3.

Differences between treatments in other skin barrier function changes were also found (Figure 4). The temperature decrease on eczematous lesions was greater after receiving dupilumab than after cyclosporine and TCS (−0.82 vs. +0.49 vs. +0.46 °C, *p* = 0.013). SCH improvement in eczematous lesions was greater in patients treated with dupilumab and TCS than in those receiving cyclosporine, while SCH increases in non-lesioned skin were higher in patients receiving dupilumab compared to TCS and cyclosporine (+11.41 vs. +0.05 vs. +6.46, *p* = 0.033). No differences in pH changes were found between treatments.

## 4. Discussion

This study shows that dupilumab is more effective than TCS and cyclosporine in improving both clinical scores, demonstrated by a higher proportion of patients achieving EASI-50, and epidermal barrier function, demonstrated by a higher proportion of participants achieving TEWL-50, after 16 weeks of treatment. 

The increased effectiveness of dupilumab in clinical practice was reflected in a higher number of patients who achieved EASI-50 and a higher reduction in SCORAD and DLQI. Clinical improvement with dupilumab was similar to that observed in previous studies in real-life settings (achieving EASI-50 in around 80% at week 12–16) [31,32,33] and higher than in clinical trials [34]. As in previous studies, we found that dupilumab improves patient quality of life [35]. Although several studies reflect the great impact of dupilumab in improving EASI, SCORAD and DLQI scores, few reports have compared its effectiveness with other systemic therapies [36]. A direct comparison between dupilumab and cyclosporine was needed, as the lack of data meant that it could not be confirmed whether these patients were really improved by dupilumab or if they could also have improved with other treatments. A network meta-analysis of randomized clinical trials showed that cyclosporine and dupilumab were similarly effective and better than placebo in improving EASI [37]. Nevertheless, another meta-analysis found that only baricitinib and dupilumab were more effective after 16 weeks of treatment than placebo while cyclosporine was not [36]. Moreover, a recent indirect comparison observed that dupilumab was more effective than cyclosporine, as manifested by a higher percentage of patients reaching EASI-50 at week 16 [38]. Our study shows that dupilumab is more effective in clinical practice, even when compared to TCS and cyclosporine.

Regarding epidermal barrier function, patients with AD are known to have higher TEWL and lower SCH in both lesioned and non-lesioned skin due to skin barrier impairment than healthy individuals [39,40]. The alkalinization of the pH in AD could also increase skin barrier dysfunction [41]. Moreover, higher temperatures could be a sign of the inflammatory changes involved in this disease [42]. Skin barrier dysfunction in AD patients has been related to proinflammatory cytokine production, mainly IL-4 and IL-13, that inhibits the expression of barrier-related molecules such as filaggrin, involucrin and loricrin, that damage the skin barrier. TCS has been reported to improve clinical inflammatory features and reduce TEWL by decreasing IL-13 production and upregulating filaggrin and loricrin expression [43]. Skin barrier improvement with dupilumab may, then, be explained by the inhibition of IL-4/IL-13 signaling, reducing markers of type 2 inflammation and reversing AD-associated epidermal abnormalities [44]. Improvements in epidermal remodeling and inflammation after dupilumab treatment have also been observed on dynamic optical coherence tomography [45].

Some other studies have also evaluated the impact of dupilumab on skin barrier function [46,47,48,49,50]. They showed that dupilumab reduced TEWL on non-involved and involved skin [46,47,48,49,50]. The impact of dupilumab in SCH differs between studies. Cristaudo et al. found reported a decreasing trend in SCH values in 30 patients with AD after 8 weeks of treatment [46] while Furuhashi et al. found that SCH did not change after 24 weeks of dupilumab treatment in seven patients [47]. We also observed that dupilumab decreases TEWL and increases SCH, reflecting skin barrier recovery. The lack of differences in SCH in other studies may be explained by the shorter follow-up [46] or the limited number of participants [47]. Moreover, we also found that dupilumab decreases temperature, which might be reflecting a reduction in the inflammatory load, while we did not find changes in pH. 

Furthermore, this is the first time to our knowledge that the impact of dupilumab on skin barrier function has been compared to other therapies, including cyclosporine and TCS. TCS also increased TEWL and decreased SCH but only in eczematous lesions. This may be because the skin compartment generates a major component of dysregulated systemic cytokines [51]. The lack of changes in skin barrier function after cyclosporine may be because this therapy does not act on the etiopathogenic axis of this disease. We also found that dupilumab is an independent factor for achieving TEWL-50. It has previously been suggested that TEWL could be a predictor for developing AD [9] or even a marker of disease severity [13] and that the decrease in TEWL after dupilumab treatment occurs mainly during the first two weeks [47], indicating that TEWL may likely be an early clinical response marker. Given the rapidly increasing number of drugs available for AD, it is important to identify markers that give an early indication of a lack of response in order to change treatment, reduce the burden of AD on patient quality of life and save healthcare costs. 

Our research is limited by the small sample size and concerns surrounding data collection: in Spain, it is compulsory to use cyclosporine before dupilumab treatment, possibly biasing the comparison between patients receiving dupilumab and cyclosporine. This bias would be toward the null hypothesis, as the real difference in skin barrier function between dupilumab and cyclosporine would be even greater than that reported in this study. The main limitation of our study that it is difficult to ensure the comparability between patients receiving each treatment. In that way, patients receiving TCS had a less severe disease that those receiving cyclosporine or dupilimab; even if the proportion of patients achieving EASI-50 in the cyclosporine group was smaller than that in the TCS group, it does not necessarily mean that TCS was better or stronger than cyclosporine. Further clinical trials should be conducted to assess the real difference in clinical and skin barrier function improvement between TCS, cyclosporin and dupilumab. Other skin barrier function parameters, such as lipid content or filaggrin, should be also measured.

## 5. Conclusions

This is the first study to evaluate the impact of dupilumab on skin barrier function and compare it to other treatments. This research could increase our understanding of the mechanism of action of dupilumab and could help clinicians select the appropriate patients to receive this treatment. Further clinical research should be conducted to determine whether patients who did not achieve TEWL-50 would subsequently fail on dupilumab and if TEWL could be considered a marker of therapeutic response in patients with AD.

## Figures and Tables

**Figure 1 jcm-11-03341-f001:**
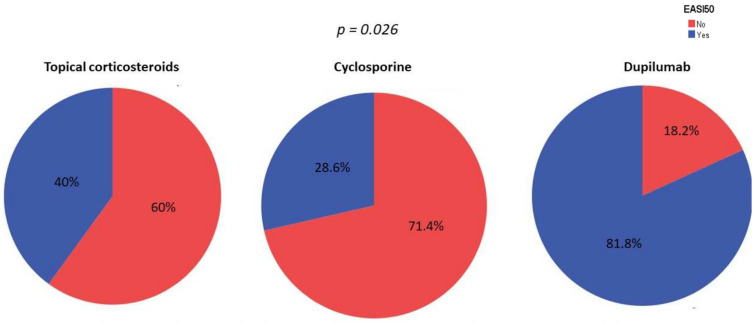
Percentage of patients achieving EASI-50 by treatment. *p*-value after conducting a multivariate logistic regression model adjusted by age, age of disease onset, sex, smoking habit and skin hydration habit.

**Figure 2 jcm-11-03341-f002:**
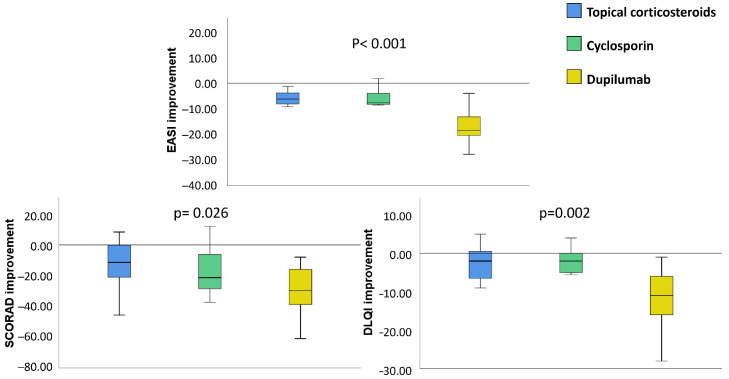
Differences in the improvement in clinical scores by treatment. DLQI, Dermatology Life Quality Index; EASI, Eczema Area Severity Index; SCORAD, SCORing Atopic Dermatitis. *p*-value after using one-way independent ANOVA test to compare changes in scores between different treatments (topical corticosteroids, cyclosporine and dupilumab).

**Figure 3 jcm-11-03341-f003:**
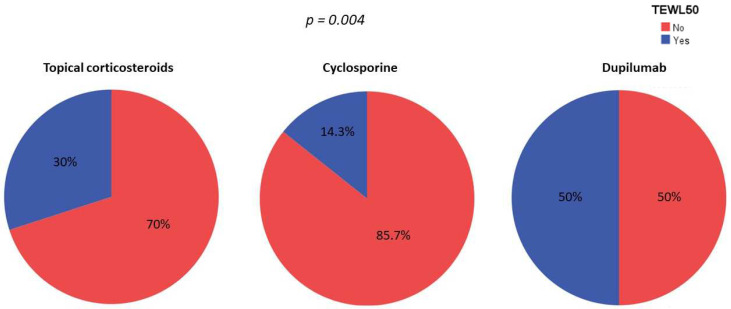
Percentage of patients reaching TEWL-50 by treatment. *p*-value after conducting a multivariable logistic regression model adjusted by age, age of disease onset, sex, smoking habit and skin hydration habit.

**Figure 4 jcm-11-03341-f004:**
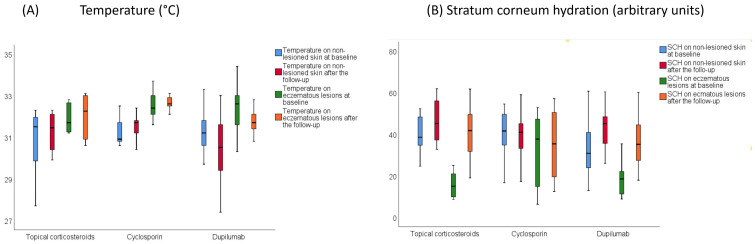
Changes in skin barrier function parameters after receiving each treatment. (**A**) Temperature (**B**) Stratum corneum hydration.

**Table 1 jcm-11-03341-t001:** Sociodemographic and clinical characteristics at baseline.

	Topical Corticosteroids (*n* = 10)	Cyclosporin (*n* = 14)	Dupilumab (*n* = 22)	*p* *
**Age (years)**	19.90 (3.76)	35.64 (22.42)	28.95 (10.83)	0.042 *
**Female sex, % (n)**	80% (8)	71.4% (10)	68.2% (15)	0.789
**Marital status**				
**- Single**	100% (10)	64.3% (9)	81.8% (18)	0.036 *
**- Married**	0	35.7% (5)	4.5% (1)	
**- Divorced**	0	0	13.6% (3)	
**Mandatory educational level (yes)**	80% (8)	71.4% (10)	88.9% (16)	0.457
**Smoking habit (yes)**	30% (3)	28.57% (4)	9.1% (2)	0.202
**Alcohol consumption (yes)**	20% (2)	42.86% (6)	27.3% (6)	0.350
**BMI (kg/m^2^)**	20.55 (0.97)	22.66 (1.71)	22.02 (3.78)	0.566
**Frecuecy of emollient use (times per week)**	5.12 (1.72)	5.25 (3.50)	6.33 (1.19)	0.378
**Skin hydration > 4 times/week (yes)**	70% (7)	92.9% (13)	72.72% (16)	0.253
**AD family history (yes)**	40% (4)	71.43% (10)	68.18% (15)	0.111
**Signs of atopic march (yes)**	60% (6)	64.29% (9)	68.18% (15)	0.835
**Disease evolution (years)**	12.20 (7.68)	8.38 (7.45)	20.67 (12.39)	0.004 *
**EASI**	11.72 (3.91)	15.04 (4.90)	24.60 (5.36)	<0.001 *
**SCORAD**	37.28 (12.11)	47.41 (9.43)	57.30 (13.83)	<0.001 *
**DLQI**	12.20 (6.20)	9.29 (4.05)	17.59 (7.28)	0.001
**BSA**	14.23 (5.31)	22.63 (7.96)	39.54 (18.47)	<0.001
**IGA**	2.5 (0.53)	3.21 (0.58)	3.73 (0.46)	<0.001

Data are expressed as mean (standard deviation) or relative (absolute) frequency. AD, atopic dermatitis; BMI, body mass index; BSA, body surface area; DLQI, Dermatology Life Quality Index; EASI, Eczema Area Severity Index; IGA, Investigator Global Assessment scale; SCORAD, SCORing Atopic Dermatitis. * *p*-value after using one-way independent ANOVA to compare continuous variables or Chi-square or Fisher test, as appropriate, to compare qualitative variables.

## Data Availability

The data presented in this study are available from the corresponding author on request.

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
