# Peer review of "Dupilumab Improves Skin Barrier Function in Adults with Atopic Dermatitis: A Prospective Observational Study"

_jcm, 2022, doi:10.3390/jcm11123341_

Round 1

Reviewer 1 Report

The manuscript reported improvement of skin barrier function in AD patients after treatment with dupilumab. I have concerns about the study.

 (Major)

Judging from the inclusion criteria described in the methods, the background of patients in TCS, cyclosporine, and dupilumab groups were totally different. It was not adequate to compare the effects of these treatment modalities if the conditions before treatment were so different. For example, even if the proportion of patients achieving EASI50 in cyclosporine group was smaller than that in TCS group, it does not necessarily mean that TCS is better or stronger than cyclosporin. The discussion has to be changed through the manuscript.

(Minor)

Skin barrier function means more than just water loss. The authors clearly showed improvement of stratum corneum hydration by dupilumab. It should be included in the abstract.

It is not clear whether TCS was allowed to use in cyclosporine and dupilumab group. How about emollients and moisturizers?

Author Response

The manuscript reported improvement of skin barrier function in AD patients after treatment with dupilumab. I have concerns about the study.

Thank you for your comments

 (Major)

Judging from the inclusion criteria described in the methods, the background of patients in TCS, cyclosporine, and dupilumab groups were totally different. It was not adequate to compare the effects of these treatment modalities if the conditions before treatment were so different. For example, even if the proportion of patients achieving EASI50 in cyclosporine group was smaller than that in TCS group, it does not necessarily mean that TCS is better or stronger than cyclosporin. The discussion has to be changed through the manuscript.

The inclusion criteria were similar between all participants. In the inclusion criteria we did not include any restriction about disease severity. Eligible patients were adults diagnosed with AD by a dermatologist according to Hanifin and Rajka criteria, 18-65 years of age, with an eczematous lesion on the volar forearm, who were scheduled to start a new treatment with TCS, cyclosporine or dupilumab. Only patients with an eczematous lesion on the volar forearm were in-cluded because the measurements were always taken in this area to homogenize differences in skin barrier function that may exist in different body areas. The exclusion criteria were: a clinical infection on the measured area, history of cancer, immunological disease or other inflammatory skin disease, incapacity to comply with the study protocol and no signed informed consent form. It is true that one of the limitations of our study is that we can’t ensure the comparability between treatments. Nevertheless, this limitation also reinforces our results because we observe higher improvement in dupilumab group when these patients had a more severe disease. We have added this information in the limitation section. Moreover, before making the comparison between groups we also compare basal vs final parameters in each group of treatment individually. We have also changed the discussion about corticosteroids and ciclosporin. We have added the following sentences: “The main limitation of our study that it is difficult to ensure the comparability between patients receiving each treatment. In that way patients receiving TCS had a less severe disease that those receiving cyclosporine or dupilimab. In that way, even if the pro-portion of patients achieving EASI-50 in cyclosporine group was smaller than that in TCS group, it does not necessarily mean that TCS was better or stronger than cyclosporine”.

(Minor)

Skin barrier function means more than just water loss. The authors clearly showed improvement of stratum corneum hydration by dupilumab. It should be included in the abstract.

We have added this sentence in the abstract: “Stratum corneum hydration increased in eczematous lesions and non-involved skin only in patients with dupilumab”

It is not clear whether TCS was allowed to use in cyclosporine and dupilumab group. How about emollients and moisturizers?

We have included information about this issue: “Patients receiving cyclosporine and dupilumab were allowed to use TCS twice a week if needed. Patients in all groups were allowed to use emollients and moisturizers when needed”.

Reviewer 2 Report

This is a very interesting article to an important topic, although in a rather small sample of patients, this makes the actual opportunities to compare between TCS only, cyclosporine and dupilumab difficult. Still the results are of interest, especially the effect of dupi on TEWL not only on lesional skin but also on uninvolved skin in atopic dermatitis, meaning a direct effect on skin barrier independent of antiinflammatory activities. The temperature story is hard to understand, here the authors should be more modest in interpretation. In the literature at least the original publication on SCORAD should be quoted. Also it would not hurt to quote a book on the disease.

Author Response

Thank you very much for all the comments

Following your recommendations, we have been more modest about the temperature results.

Moreover, we have quoted the original publication on SCORAD and also a book on the disease.

Reviewer 3 Report

The chosen topic is very interesting and of great importance in practice. The treatment of patients with atopic dermatitis must include replenishing the structure and improving the function of the skin barrier. The results obtained are promising for the newly introduced Dupilumab. Material selected correctly, methods adapted to the study design. Literature selected appropriately.

Author Response

Thank you very much for all the comments

Reviewer 4 Report

The study focuses on skin barrier dysfunction which is as a major component in pathogenesis and outcome parameter of AD. The authors have chosen a clinically relevant topic which is of interest to readers as biologicals especially dupilumab is getting more widely used in moderate severe AD.  The author has presented the materials and results systemically.  There are some points for further clarification: 

1.  TEWL, skin temperature are  "physiological" parameters of skin barrier function but less commonly referred to as "Biomarkers" Line 62.   referring to definition of biomarkers, though physiological parameters may be included by FDA but European defines "molecules" in blood or body fluid, tissues as biomarkers.  

Renert-Yuval Y, Thyssen JP, Bissonnette R, Bieber T, Kabashima K, Hijnen D, Guttman-Yassky E. Biomarkers in atopic dermatitis-a review on behalf of the International Eczema Council. J Allergy Clin Immunol. 2021 Apr;147(4):1174-1190.e1. doi: 10.1016/j.jaci.2021.01.013. Epub 2021 Jan 28. PMID: 33516871.

2. The study design is an observational study and the limitations shall be elaborated. It is unfair to compare the outcomes of the 3 groups of eczema of different severity ranging from mild to severe.  Dupulimab is indicated in moderate to severe AD, mild AD shall not be included in this study. From the presented data, the group on TCS (mild to moderate) has mean SCORAD of 35, were these mostly of moderate severity?  Further, the cyclosporin group already failed treatment with TCS and dupulimab group failed cyclosporin(and TCS too?), the 3 groups were quite different from the start. 

3. For cyclosporin group, how to define treatment failure with TCS and was TCS stopped for a period before starting treatment?  For dupulimab group, how to define failure of treatment with cyclosporin and was cyclosporin +/- TCS stopped for a period before treatment? It may be better to focus the study on patients with moderate to severe AD.  The previous treatment might have already improved the skin barrier dysfunction from the baseline and adding treatment may improve the parameters more. 

4.  Were patients given other topical or systemic treatment eg Topical calcineurin inhibitors, crisaborale ? Were TCS strictly stopped while using cyclosporin and dupulimab? 

5.  Elaborate " skin hydration for > 4 times per week" and "skin hydration habits" in more details. Use of emollients or types of emollients ?  Emollient is a important intervention that affect skin barrier function and shall be better quantified.

6. 60-70% of the patients in the TCS/ cyclosporin groups did not reach improvement in EASI 50, similarly with TEWL50.  This suggest a correlation of clinical improvement and skin barrier function.   However, concluding that "dupilumab improves skin barrier function greater than TCS or cyclosporin on both eczematous and non-eczematous lesions" will need a controlled interventional study. 

7. In addition to the references quoted (ref 44, 45) Some other studies related to dupilumab and skin barrier status are published lately 

Ferrucci S, Romagnuolo M, Alberto Maronese C, et al. Skin barrier status during dupilumab treatment in patients with severe atopic dermatitis. Therapeutic Advances in Chronic Disease. January 2021. doi:10.1177/20406223211058332

Lee SJ, Kim SE, Shin KO, Park K, Lee SE. Dupilumab Therapy Improves Stratum Corneum Hydration and Skin Dysbiosis in Patients With Atopic Dermatitis. Allergy Asthma Immunol Res. 2021 Sep;13(5):762-775. doi: 10.4168/aair.2021.13.5.762. PMID: 34486260; PMCID: PMC8419647.

Berdyshev, Evgeny, Elena Goleva, Robert Bissonnette, Irina Bronova, Anna Sofia Bronoff, Brittany Richers, Shannon Garcia, Marco Ramirez Gama, Patricia Taylor, Gabriel Bologna, Inoncent Agueusop, Mark Boguniewicz, Sivan Harel, Noah Levit, Ana Rossi, Annie Zhang, and Donald Leung. "Dupilumab Treatment Significantly Improves Skin Barrier Function in Adult and Adolescent Patients with Moderate to Severe Atopic Dermatitis." Journal of Allergy and Clinical Immunology 149.2 (2022): AB10. Web.

Overall,  it is reasonable to conclude that other than improvement in clinical scores, the physiological parameters of skin barrier dysfunctions also showed improvement during dupulimab treatment in moderate severe eczema similar to the above studies.  The comparison with TCS and cyclosporin in this study however is much limited by the study design.  Other limitations of this study include other skin barrier function  like lipid content, filaggrin not being studied shall be mentioned. 

Author Response

The study focuses on skin barrier dysfunction which is as a major component in pathogenesis and outcome parameter of AD. The authors have chosen a clinically relevant topic which is of interest to readers as biologicals especially dupilumab is getting more widely used in moderate severe AD.  The author has presented the materials and results systemically.  There are some points for further clarification: 

Thank you very much for all the comments

  1.  TEWL, skin temperature are  "physiological" parameters of skin barrier function but less commonly referred to as "Biomarkers" Line 62.   referring to definition of biomarkers, though physiological parameters may be included by FDA but European defines "molecules" in blood or body fluid, tissues as biomarkers.  

Renert-Yuval Y, Thyssen JP, Bissonnette R, Bieber T, Kabashima K, Hijnen D, Guttman-Yassky E. Biomarkers in atopic dermatitis-a review on behalf of the International Eczema Council. J Allergy Clin Immunol. 2021 Apr;147(4):1174-1190.e1. doi: 10.1016/j.jaci.2021.01.013. Epub 2021 Jan 28. PMID: 33516871.

Following your recommendations, we have changed the word biomarker to parameter.

  1. The study design is an observational study and the limitations shall be elaborated. It is unfair to compare the outcomes of the 3 groups of eczema of different severity ranging from mild to severe.  Dupulimab is indicated in moderate to severe AD, mild AD shall not be included in this study. From the presented data, the group on TCS (mild to moderate) has mean SCORAD of 35, were these mostly of moderate severity?  Further, the cyclosporin group already failed treatment with TCS and dupulimab group failed cyclosporin(and TCS too?), the 3 groups were quite different from the start. 

As mentioned in the limitation section in Spain, it is compulsory to use cyclosporine before dupilumab treatment, possibly biasing the comparison between groups. Moreover, the first-line treatment for atopic dermatitis in TCS so all patients had previously received and failed to TCS. This is one of the main limitations of our study it is difficult to ensure the comparability between group. This information has been added in the limitation section.

Patients starting dupilumab had to stop immunosuppressive agents, including cyclosporin, at least 4 weeks before dupilumab initiation. Both patients starting cyclosporine and dupilumab should not use TCS or calcineurin inhibitors at least 24 hours before baseline measurement. This information has been also added in the material and methods section.

  1. For cyclosporin group, how to define treatment failure with TCS and was TCS stopped for a period before starting treatment?  For dupulimab group, how to define failure of treatment with cyclosporin and was cyclosporin +/- TCS stopped for a period before treatment? It may be better to focus the study on patients with moderate to severe AD.  The previous treatment might have already improved the skin barrier dysfunction from the baseline and adding treatment may improve the parameters more. 

Patients starting dupilumab had to stop immunosuppressive agents, including cyclosporin, at least 4 weeks before dupilumab initiation. Both patients starting cyclosporine and dupilumab should not use TCS or calcineurin inhibitors at least 24 hours before baseline measurement. This information has been also added in the material and methods section. Cyclosporine failure was defined if no improvement was reached after 16-weeks treatment. We agree that previous treatment might have already improved the skin barrier dysfunction, so our study could be biased towards the null, meaning that the real difference between treatments would be even higher that the one reported in our study.

  1.  Were patients given other topical or systemic treatment eg Topical calcineurin inhibitors, crisaborale ? Were TCS strictly stopped while using cyclosporin and dupulimab? 

Both patients starting cyclosporine and dupilumab should not use TCS or calcineurin inhibitors at least 24 hours before baseline measurement. Patients in all groups were allowed to use emollients and moisturizers when needed but not 24 hours before the baseline measure.

  1.  Elaborate " skin hydration for > 4 times per week" and "skin hydration habits" in more details. Use of emollients or types of emollients ?  Emollient is a important intervention that affect skin barrier function and shall be better quantified.

Patients were asked about the frequency they used emollients per week. Patients receiving TCS used them 5.12 (1.72) times per week, patients receiving cyclosporine 5.25 (3.50) and those receiving dupilumab 6.33 (1.19) times per week without significant differences between groups (p=0.378). This information has been added in the table 1.

  1. 60-70% of the patients in the TCS/ cyclosporin groups did not reach improvement in EASI 50, similarly with TEWL50.  This suggest a correlation of clinical improvement and skin barrier function.   However, concluding that "dupilumab improves skin barrier function greater than TCS or cyclosporin on both eczematous and non-eczematous lesions" will need a controlled interventional study. 

After conducting a logistic regression model, adjusted by possible confounding factors, we observed that dupilumab was an independent factor for reaching EASI-50 and TEWL-50. We agree that the most proper study to find differences between treatments is a clinical trial. Following your recommendations, we have added the following sentence in the discussion: Further clinical trials should be conducted to assess the real difference in clinical and skin barrier function improvement between TCS, cyclosporin and dupilumab.

  1. In addition to the references quoted (ref 44, 45) Some other studies related to dupilumab and skin barrier status are published lately 

Ferrucci S, Romagnuolo M, Alberto Maronese C, et al. Skin barrier status during dupilumab treatment in patients with severe atopic dermatitis. Therapeutic Advances in Chronic Disease. January 2021. doi:10.1177/20406223211058332

Lee SJ, Kim SE, Shin KO, Park K, Lee SE. Dupilumab Therapy Improves Stratum Corneum Hydration and Skin Dysbiosis in Patients With Atopic Dermatitis. Allergy Asthma Immunol Res. 2021 Sep;13(5):762-775. doi: 10.4168/aair.2021.13.5.762. PMID: 34486260; PMCID: PMC8419647.

Berdyshev, Evgeny, Elena Goleva, Robert Bissonnette, Irina Bronova, Anna Sofia Bronoff, Brittany Richers, Shannon Garcia, Marco Ramirez Gama, Patricia Taylor, Gabriel Bologna, Inoncent Agueusop, Mark Boguniewicz, Sivan Harel, Noah Levit, Ana Rossi, Annie Zhang, and Donald Leung. "Dupilumab Treatment Significantly Improves Skin Barrier Function in Adult and Adolescent Patients with Moderate to Severe Atopic Dermatitis." Journal of Allergy and Clinical Immunology 149.2 (2022): AB10. Web.

Overall,  it is reasonable to conclude that other than improvement in clinical scores, the physiological parameters of skin barrier dysfunctions also showed improvement during dupulimab treatment in moderate severe eczema similar to the above studies.  The comparison with TCS and cyclosporin in this study however is much limited by the study design.  Other limitations of this study include other skin barrier function  like lipid content, filaggrin not being studied shall be mentioned. 

Following your recommendations, we have added this information as a limitation of our study. Moreover, we have added the suggested references

Round 2

Reviewer 1 Report

The manuscript has been adequately revised.